# Endothelial Semaphorin 3F Maintains Endothelial Barrier Function and Inhibits Monocyte Migration

**DOI:** 10.3390/ijms21041471

**Published:** 2020-02-21

**Authors:** Huayu Zhang, Dianne Vreeken, Abidemi Junaid, Gangqi Wang, Wendy M. P. J. Sol, Ruben G. de Bruin, Anton Jan van Zonneveld, Janine M. van Gils

**Affiliations:** Einthoven Laboratory for Vascular and Regenerative Medicine, Department of Internal Medicine, Leiden University Medical Center, 2333 ZA Leiden, The Netherlandsg.wang@lumc.nl (G.W.); rgdebruin@gmail.com (R.G.d.B.); A.J.van_Zonneveld@lumc.nl (A.J.v.Z.)

**Keywords:** semaphorin, endothelial cells, monocytes

## Abstract

In normal physiology, endothelial cells (ECs) form a vital barrier between the blood and underlying tissue controlling leukocyte diapedesis and vascular inflammation. Emerging data suggest that neuronal guidance cues, typically expressed during development, have roles outside the nervous system in vascular biology and immune responses. In particular, Class III semaphorins have been reported to affect EC migration and angiogenesis. While ECs express high levels of semaphorin 3F (SEMA3F), little is known about its function in mature ECs. Here we show that SEMA3F expression is reduced by inflammatory stimuli and increased by laminar flow. Endothelial cells exposed to laminar flow secrete SEMA3F, which subsequently binds to heparan sulfates on the surface of ECs. However, under pro-inflammatory conditions, reduced levels of SEMA3F make ECs more prone to monocyte diapedesis and display impaired barrier function as measured with an electric cell–substrate impedance sensing system and a microfluidic system. In addition, we demonstrate that SEMA3F can directly inhibit the migration of activated monocytes. Taken together, our data suggest an important homeostatic function for EC-expressed SEMA3F, serving as a mediator of endothelial quiescence.

## 1. Introduction

Endothelial cells (ECs) form a vital barrier between the blood and underlying tissue, and are crucial for maintaining vessel homeostasis by controlling vascular tone, blood fluidity, endothelial permeability and the regulation of plasma constituents and leukocyte transendothelial migration [1]. The observation that the anatomy and the cellular transcriptomes of the vascular systems of vertebrates often overlap with that of their nervous systems has led to one of the major recent insights in developmental vascular biology, i.e., that the coordinated patterning of nerves and vessels is achieved by each system separately using the same cues and signals [2,3]. These conserved patterning factors, together called the neuronal guidance cues, were first identified in neural development and involve four major families of conserved ligands being the semaphorins (SEMAs), netrins, ephrins and slits. Neuronal guidance cues act together through a complex interplay of short and long-range signals that can either repel or attract the cells of the developing network [4]. More recently, their critical roles in physiological and pathological regulation of vascular biology and immune responses have been increasingly recognized.

In particular, members of the Class III SEMA family have been reported to affect endothelial biology [5,6,7]. Class III SEMAs are secreted proteins typically composed of a signal peptide for cellular secretion, a SEMA domain (involved in dimerization and receptor binding to plexins/neuropilins), an Immunoglobulin (Ig) domain, a PSI domain (domain found in Plexins, Semaphorins and Integrins necessary for receptor binding to neuropilins) and a basic tail [8]. Class III SEMA protein molecules form homo-dimers by disulfide bonds and bind to its canonical receptors Plexin (PLXN) A1-4 or D1 with the help of neuropilin (NRP) 1 or 2, which increases ligand binding affinity [9]. The intracellular regions of these PLXN receptors have a Rho-GTPase binding domains and an R-Ras GTPase activating protein domain, allowing the interaction with small GTPases [10,11]. Of the Class III SEMAs the function of SEMA3A has been investigated the most in endothelial biology. Less is known about the necessity of SEMA3F in EC biology, though SEMA3A has only moderate expression and almost all types of ECs express high levels of SEMA3F [6].

In this article, we confirmed the high expression of SEMA3F in ECs and demonstrated its modulation by shear stress and inflammatory factors. We further investigated the functional importance of SEMA3F and propose the involvement of SEMA3F in endothelial barrier function and monocyte migration. 

## 2. Results

### 2.1. Semaphorin 3F Is the Highest Expressed Class III Semaphorin in Endothelial Cells

We set off by studying the expression of Class III SEMAs in ECs using available transcriptomics data on GPL570 platform, a commonly used human transcriptome microarray platform. Of the Class III SEMAs, SEMA3A, SEMA3C and SEMA3G are abundantly expressed by some of the ECs (Figure 1A). High abundance of SEMA3F transcripts was detected in all types of ECs profiled to date. The SEMA3F levels were in several types of ECs comparable to that of vascular endothelial cadherin (VE-cadherin, CDH5), the main component of adherent junctions. Epithelial cadherin (E-cadherin, CDH1) was taken along as negative control. Endothelial cells from larger arteries, such as aortic ECs, coronary artery ECs, iliac artery ECs and umbilical artery ECs, showed higher expression of SEMA3F, compared to microvascular ECs. In order to compare SEMA3F expression level of the ECs to other cell types, all 369 cell types that were profiled on GPL570 platforms were ranked by expression of SEMA3F. The top 10 cell types consist of five EC types and five epithelial-related cell types (Figure 1B).

### 2.2. Expression of SEMA3F Is Regulated by Inflammatory Cytokines and Shear Stress

Given the high expression of SEMA3F in ECs, we next assessed if inflammatory and hemodynamic conditions affected the expression of SEMA3F. For this we used human umbilical vein endothelial cells (HUVECs), in which we confirmed the expression of Class III SEMAs by qPCR (Figure 2A). Treatment with inflammatory cytokines decreased the expression of SEMA3F (Figure 2B). HUVECs treated with tumor necrosis factor alpha (TNFα) had a decrease in expression by 50% after 6 hours and the effect persist until at least 24 hours. Similar observations were made for treatment with interleukin 1 beta (IL1β) after 6 hours, while the expression of SEMA3F was restored after 24 hours of IL1β treatment. The expression of SEMA3A and SEMA3G were changed upon inflammatory cytokine stimulation, however only IL1β reduced SEMA3A expression, while IL1β increased SEMA3G levels, as did TNFα for SEMA3A expression upon 24 h stimulation (Appendix A).

To study the regulation of SEMA3F expression by shear stress, ECs were cultured under laminar flow or oscillatory flow (10 dyn/cm^2^) for 1 day or 7 days and gene expression to static cultured ECs was compared. As expected, laminar shear stress induced the expression of shear-dependent transcription factor Kruppel Like Factor 2 (KLF2) after 1 day and further after 7 days, while oscillatory shear stress failed to do so (Figure 3A). Quantitative PCR revealed that mRNA expression of SEMA3F was also augmented by a shear dependent mechanism, for SEMA3A and SEMA3G this was not observed (Appendix A). The expression of SEMA3F followed a similar pattern as KLF2 with a significant 2.4-fold increased expression after 7 days (Figure 3B). Moreover, immunoblot analysis of control (static) and 7 days sheared ECs revealed an evident increase in SEMA3F protein expression, as well as known shear responsive NOS3 (Figure 3C). Interestingly, presence of SEMA3F dimers could only be detected in the conditioned medium from ECs cultured under laminar flow, but not in the static condition (Figure 3C). This implied that shear stress not only increased the transcription and translation of SEMA3F, but also the secretion of SEMA3F protein. Immunofluorescence staining of SEMA3F revealed colocalization with heparan sulfate on the surface of the cells (Figure 3D–F). Treatment with heparanase reduced the presence of SEMA3F on the flow cultured ECs (Figure 3D–F), indicating binding of SEMA3F to the endothelial glycocalyx, a network of membrane-bound proteoglycans and glycoproteins. 

To assess if SEMA3F expression can be mediated by the shear induced transcription factor KLF2, we overexpressed KLF2 in ECs (Figure 4A). Overexpression of KLF2 resulted in increased expression of SEMA3F mRNA and protein in static cultured cells (Figure 4B,C). To confirm the KLF2-dependent regulation of SEMA3F under laminar shear stress we silenced KLF2 in ECs using shRNA vectors, and cultured these cells for 7 days under flow (Figure 4D). Knockdown of KLF2 in endothelial cells suppressed the induction of SEMA3F by laminar flow (Figure 4E). NOS3 detection was used to validate functional overexpression or knock down of KLF2 (Figure 4C,F).

### 2.3. Loss of SEMA3F Impairs Endothelial Barrier Function

In order to study the functional importance of SEMA3F for ECs, SEMA3F expression in ECs was silenced using lentiviral shRNA vectors targeting the SEMA3F mRNA. A non-SEMA3F targeting scrambled shRNA was used as a control. Repression of SEMA3F (SEMA3F KD) was validated using qPCR (Figure 5A). Endothelial cells with reduced levels of SEMA3F showed a more spindle-like and elongated morphology compared to scrambled shRNA transduced cells (Figure 5B,C). Immunostaining of CDH5 revealed striped discontinuous adherens junctions (Figure 5C,D), which is often caused by excessive pulling force from cytoskeleton to the junction [12]. Indeed, staining for filamented actin (F-actin) showed a surfeit of stress fiber (Figure 5C,E).

The morphological change strongly indicates the possibility of impaired endothelial barrier function after reduction of SEMA3F expression. To test this postulation we assessed the effect of SEMA3F KD on endothelial barrier function through the use of electric cell–substrate impedance sensing (ECIS) [13] and the more physiological 3D-endothelial tube in a microfluidics platform [14]. With each method we found that, targeted reduction of SEMA3F in ECs resulted in impaired barrier function (Figure 6A–C). The impaired barrier function of cells with SEMA3F KD could be improved by treating the endothelial cells with supernatant of non-transduced endothelial cells cultured under laminar flow, indicated by more localization of CDH5 at cell–cell contacts and increased impedance measurement (Figure 6D,E).

Next, we assessed whether loss of barrier function upon SEMA3F knock down alters monocyte transendothelial migration with monocyte chemotaxis over Transwell filters containing endothelial monolayers with SEMA3F knock down or control cells. Consistent with the reduced barrier function, we found that loss of SEMA3F led to increased monocyte transendothelial migration (Figure 6F).

### 2.4. SEMA3F Itself Works as an Inhibitor of Migration for Activated Monocytes

Since SEMA3A can inhibit directional migration of monocytes [15], we investigated whether SEMA3F could also affect monocyte migration by directly acting on monocytes. For this, we started by looking at the expression of SEMA3F receptors, namely PLXNA1 and PLXNA3, and the co-receptor NRP2, by naïve monocytes and inflammatory stimulated monocytes. We found that monocytes express all these receptors, and their expression levels are induced upon stimulation with TNFα (Figure 7A). Next, SEMA3F was either used to pretreat the monocytes or it was added to the Transwell migration system, either in the bottom or in both compartments. Consistent with our receptor expression data, we observed that the migration of TNFα-stimulated monocytes was inhibited by either pretreatment of SEMA3F or addition of SEMA3F to the monocytes during the migration experiment. Having SEMA3F only in the bottom compartment of the Transwell system did not affect migration (Figure 7B,C). Together these observations have demonstrated that SEMA3F does have an inhibitory effect on migration inflammatory stimulated monocyte, but not a repellent effect. 

## 3. Discussion

In this study, we confirmed the high expression of SEMA3F in ECs found with gene expression meta-analysis of omni datasets. We revealed a KLF2-dependent regulation of SEMA3F under laminar shear stress. Using an shRNA knock down model, we pinpointed an essential role for SEMA3F in maintaining the endothelial barrier function and inhibiting the mobility of monocytes. Collectively, these results support that SEMA3F expressed by ECs can serve as a mediator of quiescent ECs. 

The function of Class III SEMAs in EC biology has received a good amount of attention with SEMA3A being the most thoroughly investigated. SEMA3A binds to PLXNA2 (instead of PLXNA1 and PLXNA3 in the case of SEMA3F), which activates the Rho-family small GTPase Rnd1 and induces a collapse of the cytoskeleton [11,16]. The ligand–receptor interaction was shown to inhibit stress fiber formation and collapse of the ECs cell membrane [17]. SEMA3A was also shown to cause internalization of CDH5, thus affecting the integrity of adherens junctions [18]. Via these mechanisms, SEMA3A increased permeability of the endothelium in several situations [18,19,20,21]. Compared to the effect of SEMA3A on endothelium permeability, available evidences for a role of SEMA3F are rather ambiguous. High doses of exogenous SEMA3F were shown to collapse the EC membrane and cytoskeleton, but it did not affect EC permeability [17]. Peritoneal injection of SEMA3F in mice caused elevation of leukocyte numbers in the peritoneal cavity, which may be explained by the disturbance of the endothelial or epithelial cell function by SEMA3F [22].

Instead of using exogenous SEMA3F, we studied the effect of reduced level of SEMA3F on EC barrier function, which should be more physiologically relevant. In line with SEMA3F being shown to inhibit stress fiber formation [23,24], after the knockdown of SEMA3F, we observed excessive amounts of F-actin, especially at the cortical regions of HUVECs. The end of stress fibers connects components of adherens junctions. Excessive amounts of stress fibers can apply a pulling force on endothelial adherens junctions, resulting in disrupted integrity of these junctions and increased permeability [25,26]. Indeed, our results show the disruption of adherens junctions with CDH5 immunofluorescence staining. The fact that SEMA3F is a shear inducible protein, it is possible that SEMA3F works as a shear-dependent auto- or paracrine regulator of endothelial cortical F-actin via PLXN-small GTPase signaling, thus stabilizing the adherens junctions. Knowledge on the exact small GTPase governing the downstream signaling is still lacking and requires further research.

One of the most common reasons for endothelial dysfunction is an adverse hemodynamic environment, evidenced by the fact that more atherosclerotic plaques can be found at bifurcations of vessels and regions with low vessel wall shear stress [27,28]. The underlying mechanism is that high (and laminar) shear stress provokes quiescence of ECs via mechano-sensing proteins and downstream shear-dependent transcription factors, including KLF2 and KLF4. Quiescent ECs have a good bioavailability of nitric oxide, better barrier function and little inflammatory responses. Low shear stress or disturbed flow (also called turbulence) stimulates EC proliferation, apoptosis and inflammation [29]. Adaption and alteration of EC function under adverse hemodynamic environments involves and induces changes in gene expression and is consequently the driver of functionalities changes [25]. For SEMA3A it has been shown, in vitro and in vivo, that its expression is decreased by ECs at regions of disturbed flow and upon inflammatory stimulations [15]. For SEMA3F, in line with the findings in our prolonged flow culture and KLF2 overexpression experiments, upregulation has been reported in a microarray of ECs overexpressing the flow-responsive transcription factor KLF2 [26]. Next to an increased SEMA3F expression upon laminar flow we observed a decrease in expression upon inflammatory stimulations in the ECs. In tumor cells it has been shown that the transcription repressors zinc finger E-box binding homeobox (ZEB)-1 and inhibitors of DNA binding/differentiation2 (Id2) proteins can decrease the expression of SEMA3F [30,31]. In tumor cells these transcription factors can be activated by inflammation [32,33], and similar pathways might be at play in ECs in regulating SEMA3F expression. Interestingly, we could not detect the SEMA3F protein in the medium of ECs cultured under static conditions, whereas the SEMA3F protein could be detected in the conditioned medium of ECs cultured under prolonged laminar flow. These observations indicated that shear stress not only coordinated induction of SEMA3F gene expression, but also stimulated secretion of SEMA3F, confirming the concept that induction of SEMA3F expression is at least a concomitant event in endothelial maturation and quiescence.

One key aspect of endothelial function is the ability to synthesize and maintain a network of membrane-bound proteoglycans and glycoproteins at the endothelium lumen, called the glycocalyx [34]. This glycocalyx contains heparan sulfate proteoglycans providing binding for endothelium- and plasma-derived soluble molecules facilitating protein–receptor interactions and the formation of surface gradients of chemokines and growth factors [35]. While heparan sulfates are shown to be essential for the development of the nervous system and specifically can affect several neuronal guidance cues [36], a role for these heparan sulfate in SEMA-mediated function is barely described. A role for heparan sulfate proteoglycans in SEMA function is suggested from the phenotypic similarities between central nervous system development in mice defective in secreted SEMA signaling and mice lacking the heparan sulfate elongation gene 1 (Ext1) [36]. For SEMA5A, an interaction with axonally expressed heparan sulfate proteoglycans has been shown [37]. In addition, De Wit et al. have shown that heparan sulfates can effectively compete SEMA3A from neuronal cells [38]. We have found that treatment with heparanase reduces the presence of SEMA3F on ECs, proving an interaction of SEMA3F with the endothelial glycocalyx probably mediated by heparan sulfate. In this way, SEMA3F might be presented to leukocytes in the circulation and/or be beneficial for a tight endothelial barrier by competing for VEGF on the glycocalyx.

In regard to leukocyte guidance, several members of the Class III SEMAs have been studied in the context of inflammation and leukocyte recruitment. Studies have revealed that these Class III SEMAs can dampen inflammation or mediate resolution of inflammation [39,40,41,42,43]. SEMA3A has been shown to inhibit migration of monocytes [15,42] and for SEMA3E it has been shown that it can inhibit the migration of macrophages [44] and neutrophils [43]. Only for mature dendritic cells there have been reports that SEMA3A, -C and -F can promote their migration [45]. For SEMA3F the functional effect on leukocyte migration had not been fully studied. In our experiments, we showed that SEMA3F could affect monocyte migration via both modulating EC adherens junction stability and acting directly on monocytes. Intriguingly, in a recent study, Reichert et al. showed that SEMA3F promotes transmigration of peripheral blood mononuclear cells via increasing PECAM1 at cellular junctions [22]. Similarly, we observed reduction of PECAM1 with reduction of SEMA3F (unpublished data). In both cases, the presence of PECAM1 in endothelial adherens junction was positively correlated with SEMA3F expression. PECAM1 is essential in maintaining endothelial barrier function and promoting monocyte diapedesis through junctions of ECs [46,47,48]. When endothelial junctions are intact, PECAM1 concentrates at the junctions, binds homophilically to leukocyte expressed PECAM1 and facilitates transmigration of these cells [49]. When endothelial junctions are disrupted, junctional PECAM1 disappears and transmigration of leukocytes is then independent of PECAM1. The above mechanism could explain why an increase of monocyte migration through the EC monolayer is observed both with the SEMA3F treatment or SEMA3F KD [22]. We also investigated whether SEMA3F could directly act on monocyte migration. To our surprise, SEMA3F did not regulate the migration of resting THP-1 cells. However, we saw that monocyte expression of SEMA3F receptors, namely NRP2, PLXNA1 and PLXNA3, were induced by stimulation with TNFα. Similar observations have been made in DCs and natural killer cells (NKs), in which SEMA receptors were upregulated upon cell activation [39,45]. We next demonstrated that migration of TNFα-stimulated monocytes was reduced by pretreatment with SEMA3F, or by having SEMA3F in the migration system, but not by having SEMA3F only in the bottom chamber of the Transwell system. Therefore, we concluded from our studies that SEMA3F acts as a general inhibitor of monocyte migration, as observed before for SEMA3A [15]. Moreover, we revealed that TNFα could sensitize monocytes by inducing the expression of SEMA3F receptors.

In conclusion, we demonstrated a novel function of SEMA3F in endothelial quiescence. As a shear-inducible secretory factor, SEMA3F is essential for the endothelial barrier function and is able to regulate monocyte migration. Together these studies provide novel insight into the role of endothelial expressed SEMA3F as a mediator of endothelial homeostasis contributing to vascular health.

## 4. Materials and Methods

### 4.1. Access of Gene Expression Omnibus Data

Curated gene expression data of all human cell types that were profiled on Affymetrix Human Genome U133 Plus 2.0 Array platform (Accession: GPL570/HG-U133_Plus_2) were obtained using Genevestigator software [50]. For each cell type, average expression value and the standard deviation were calculated. Graphs were created with R package ggplot2 [51].

### 4.2. Cell Culture

HEK293T cells (ATCC, CRL-3216) were cultured in DMEM (Gibco) supplemented with 10% FCS (*v*/*v*) and antibiotics. THP-1 cells (ATCC, TIB-202) were cultured in RPMI 1640 (Gibco, Paisley, UK) supplemented with 10% FCS (*v*/*v*), antibiotics and 0.05 mM 2-mercaptoethanol. Primary HUVECs were isolated from in house human umbilical cords and were cultured in EGM2 medium (Lonza, Walkersville, MD, USA) supplemented with antibiotics. Gelatin (0.5% *w*/*v* in water) coated plates or flasks were used for HUVECs. All experiments using HUVECs were repeated at least 3 times using cells from different donors. All cells were maintained in a 37 °C incubator with 5% CO_2_.

### 4.3. Laminar or Oscillatory Shear Stress Culture Conditions

Culturing cells under laminar or oscillatory flow was performed using an iBiDi flow system according to the manufacture’s instruction. Briefly, HUVECs were seeded into closed perfusion chambers (IbidiTreat 0.4 µ-Slide I, Luer, Ibidi, Martinsried, Germany) and allowed to adhere for 15 minutes. The 2 inlets of the slide were then connected to 2 syringes, which acts as the reservoir for medium. The tops of the 2 syringes were connected to an iBiDi pump, which introduces pressure on one of the syringes in alternate manner. A central valve was placed on the tube, which allowed alternation of the flow direction. For the laminar flow condition, the alternation of pressure introduction was synchronized with the valve-controlled alternation of flow direction, resulting in unidirectional flow over the cells with constant shear stress. For oscillatory flow, the pump still alternated similar to laminar flow, while the valve alternated every 1 second, creating oscillatory flow over the cells. The pump setup allowed perfusion of culture medium over the monolayer of ECs at a shear stress of 10 dyn/cm^2^, both laminar and oscillatory. The chamber and reservoirs containing the medium were kept in an incubator at 37 °C and 5% CO_2_. Medium was refreshed after 1 and 4 days of culture. After 1 day or 7 days of flow, tubing connections were removed. Trizol (Invitrogen, 15596026, Carlsbad, CA, USA) or RIPA (Cell Signaling Technology, #9806, Danvers, MA, USA) buffer were added for RNA analysis or protein analysis, respectively.

### 4.4. Lentiviral Vector for KLF2 Overexpression or SEMA3F or KLF2 Knockdown

Second generation lentiviral system was used for delivery of a plasmid containing the KLF2 gene or a plasmid containing short hairpin RNA (shRNA) construct targeting SEMA3F mRNA (SEMA3F KD) or KLF2 mRNA (KLF2 KD). Plasmids without the KLF2 gene (mock) or with a scrambled shRNA (scrambled) were used as controls. Briefly, packaging plasmids (pPAX), envelope plasmids (pVSVG) and plasmids containing shRNA constructs were transfected into HEK293T cells using the polyethylenimine transfection method. The culture medium containing lentivirus particles was harvested 2 days and 3 days after transfection and was filtered with 0.45 µM filters to exclude cell remnants. The lentiviral particle containing medium was then used to transduce HUVECs with dilution ratio of 1:10 in EGM2 medium. The inoculum was removed after 1 day. Puromycin (2 µg/mL, Thermo Fisher, Grand Island, NY, USA) selection was done 2 days after the lentiviral transduction. Cells that were resistant to puromycin for 24 hours were subcultured and used in subsequent experiments.

### 4.5. Quantitative Polymerase Chain Reaction

For quantification of mRNA expression, cells were lysed in Trizol (Invitrogen, 15596026, Carlsbad, CA, USA) and total RNA was isolated using RNeasy miniKit (Qiagen, 74106, Hilden, Germany) according to the manufacturer’s instructions. First strand complementary DNA synthesis was done using M-MLV reverse transcriptase kit (Promega, M1701, Madison, WI, USA) with oligodT as primer. Quantitative polymerase chain reactions (qPCR) were done using SYBR green/polymerase mix (Applied Biosystems, 4472903, Vilnius, Lithuania) on a Biorad CFX384 system. GAPDH served as the reference gene. List of primers used in qPCR is provided in Table 1.

### 4.6. Western Blot Analysis

For Western blot analysis, cells were lysed with the ice cold RIPA buffer (Cell Signaling Technology, #9806, Danvers, MA, USA) supplemented with a Complete Protease Inhibitor Cocktail (Roche, 11836153001, Mannheim, Germany). The cell debris was precipitated by centrifuging at 14,000 rpm for 20 min and supernatant was harvested. Protein concentration in samples was measured using Pierce BCA Protein Assay Kit (ThermoFisher Scientific, 23225, Rockford, IL, USA). For electrophoresis, 10 µg samples in 20 µL were prepared with loading buffer (Cell Signaling Technology, #7722, Danvers, MA, USA) with or without DTT, depending on whether the reductive environment was needed. After electrophoresis separation on a 4–15% polyacrylamide gel (BioRad, 4561084, Temse, Belgium), proteins on gels were transferred onto PVDF membranes (BioRad, 1704156, Temse, Belgium) using BioRad turbo transfer system (BioRad, 1704150). The membranes were blocked using 5% BSA solution (*w*/*v* in TBST). The primary antibodies were prepared in 5% BSA solution: rabbit anti-human SEMA3F (Abcam, ab135880 1:500, Cambridge, UK) and rabbit anti-human GAPDH (Cell Signaling Technology, #2118, 1:5000, Danvers, MA, USA). After overnight incubation of primary antibody at 4 °C, secondary antibody (Dako, P044801-2, 1:5000, Glostrup, Denmark) was added and incubated for 1 hour. HPR signal was detected with a Pierce™ ECL Western Blotting Substrate (ThermoFisher Scientific, 34095, Rockford, IL, USA) and captured by the Chemidoc system (BioRad, 17001401). The images were analyzed in ImageJ software. 

### 4.7. Immunofluorescence Staining

For immunofluorescence staining, HUVECs were cultured on µ-Slide 8 well (Ibidi, C80826, Martinsried, Germandy) and fixated with 4% formaldehyde (*v*/*v* in HBSS supplemented with calcium and magnesium (HBSS+), Gibco, Paisley, UK) for 10 minutes. Cells were then permeabilized with 0.1% triton-X100 for 2 minutes for F-actin and blocked by 5% BSA (*w*/*w* in HBSS) for 30 minutes. Except when staining for heparan sulfate and SEMA3F, then cells were not permeabilized and directly incubated with blocking solution. The primary antibody was prepared in 0.5% BSA (*w*/*v* in HBSS+): CDH5 mouse anti-human antibody (BD Biosciences, 555661, 1:200, San Jose, CA, USA), SEMA3F rabbit anti-human antibody (Abcam, ab77760, 1:50, Cambridge, UK) and heparan sulfate (JM403; 1.1 mg/mL, gift from Dr. van der Vlag and Dr. Kuppevelt (Nijmegen Center for Molecular Life Sciences, Radboud University Medical Center, Nijmegen, The Netherlands), 1:110). Overnight incubation of the primary antibody was done at 4 °C. The secondary antibody was also prepared in 0.5% BSA (goat anti-mouse IgG alexa 488, goat anti-rabbit IgG alexa 568 or goat anti-mouse alexa 488, Invitrogen, 1:250, Eugene, OR, USA), together with Rhodamine Phalloidin (Invitrogen, 1:200, Eugene, OR, USA) and Hoechst (Invitrogen, 1:2000, Eugene, OR, USA) to stain F-actin and cell nuclei. Fluorescence images were captured by the Leica sp5 confocal microscopic system and analyzed using ImageJ software.

### 4.8. Electric Cell–Substrate Impedance Sensing

Electric cell–substrate impedance sensing (ECIS) assays were done using the ECIS Ztheta (Applied BioPhysics, Troy, NY, USA) device with the standard 8 well arrays (Applied BioPhysics, 8W10E) or 96 wells array (Applied BioPhysics, 96W20idf), which measures the change in the EC monolayer barrier function in real time [52]. The plate was treated with 10 mM l-cysteine for 10 minutes to prepare the surface of gold electrodes and washed twice with water. We then coated the plate with 0.5% gelatin solution for 30 minutes at 37 °C. After the coating, the gelatin solution was washed away with the EGM2 medium and 400 µL medium was added to each well. Thereafter, cell-free baseline measurements were taken. After approximately 30 minutes, measurements were paused and 1 × 10^5^ HUVECs in suspension (200 µL) were added to the wells by mixing the cell suspension with medium in the wells. The measuring was then resumed for at least 30 hours. For both baseline and experiment timepoints, impedance was measured at multiple frequencies every 5 minutes. Measurements at 4000 Hz were used for data analysis, because impendence is contributed mainly from cell junction resistance at this frequency.

### 4.9. Permeability of 3D Endothelial Culture

HUVECs with or without SEMA3F knockdown were seeded in gelatin coated microvascular channels of custom-made gradient design OrganoPlate^®^ (Mimetas, Leiden, the Netherlands) with 4 mg/mL type 1 collagen (Trevigen; Gaithersburg, United States) in the extracellular matrix channel. After allowing the cells to adhere for 1 hour, culture medium was replace by a mix of Endothelial Cell Growth Medium MV2 (PromoCell, Heidelberg, Germany) and Pericyte Growth Medium (Angio-Proteomie, Boston, MA, USA) in a ratio of 1:1. The device was placed on a rocker platform with 7° angle of motion and eight minutes timed operation to allow continuous flow of medium in the microvessels. After 24 hours, the medium was refreshed and the HUVECs were cultured for 3-4 more days. To measure vessel permeability, the culture medium in the extracellular matrix (ECM) channels was replaced by HBSS+ buffers and the culture medium of the microvascular channels were replaced with HBSS+ buffer containing 125 µg/mL Albumin-Alexa 555 (Life Technologies, Eugene, OR, USA). Following this, the OrganoPlate^®^ was placed in the environmental chamber (37 °C; 5% CO_2_) of a fluorescent microscope system (Nikon Eclipse Ti) and time-lapse images were captured. The permeability coefficient was calculated by determining the fluorescent intensities in the microvascular channel and in the ECM channel of the capture images. The fluorescent intensity of the ECM channel was normalized with the fluorescent intensities in the microvascular channel of each measured time point. This showed the change in intensity ratio inside the gel channel as a function of time. The scatter plot was fitted with a linear trend line to determine the slope. Finally, using Fick’s First Law the apparent permeability was determined as:
$$P_{app}\left(\cdot 10^{-6}\;cm/s\right)=\frac{d\left(\frac{I_{g}}{I_{p}}\right)}{dt}\cdot \frac{A_{g}}{l_{w}}$$
where $I_{p}$ is the intensity in the microvascular channel, $I_{g}$ is the intensity in the ECM channel,
$A_{g}$
$\left(480\cdot 10^{-6}\;cm^{2}\right)$ is the area of the ECM channel and
$l_{w}$
$\left(400\cdot 10^{-4}\;cm\right)$ is the length of the vessel wall that separates between ECM and microvascular region. All data analysis was done with ImageJ (http://rsbweb.nih.gov/ij/, NIH, United States) and Matlab (MathWorks).


### 4.10. Transwell Migration

Chemotaxis of THP-1 monocytes was measured using 24-well Transwell culture inserts with a 5 μm pore size filter (Corning, 734-1573, Kennebunk, ME, USA), either coated with 10 µg/mL Fibronectin (Sigma, F4759, St. Louis, MO, USA) or containing an endothelial monolayer. For transendothelial migration experiments, ECs were seeded at density of 4 × 10^4^ cells/cm^2^ on the Transwell insert one day before the experiments and then kept in EGM2 (100 µL in the top chamber and 500 µL in the bottom chamber). Two hours before migration, EGM2 was replaced by migration medium (RPMI 1640 with 0.25% BSA) to let ECs adapt to the new medium. When indicated, 15 h before the experiment, THP-1 cells were treated with 2 ng/mL TNFα. Before the experiment, THP-1 cells were washed with migration medium and resuspended to a concentration of 1 × 10^7^ per mL. Calcein AM (ThermoFisher, C3100MP, 1:1000, Eugene, OR, USA) was added so that THP-1 cells would carry green fluorescence. After labeling, THP-1 cells were washed twice in migration medium and resuspended to a concentration of 1 × 10^6^/mL. For SEMA3F pretreatment, 500 ng/mL SEMA3F (gift from Dr. Nakayama and Prof. Briscoe, Boston Children’s Hospital and Harvard Medical School, Boston, MA, USA) was added to the suspension and cells were incubated for 30 minutes. After 30 minutes, THP-1 cells were added to Transwell inserts and the inserts were placed on wells with 500 µL migration medium supplemented with 10 nM MCP1 (R&D Systems, 279-MC, Minneapolis, MN, USA) with or without 500 ng/mL SEMA3F. The cells were allowed to migration for 2 hours. Thereafter, the cells in the bottom chambers (containing migrated THP-1 cells) were pelleted and lysed by 1% TritonX-100. Fluorescent signal of Calcein AM was measured by a colorimetric analyzer (excitation: 480 nm; emission: 525 nm), representing the number of THP-1 cells migrated.

## Figures and Tables

**Figure 1 ijms-21-01471-f001:**
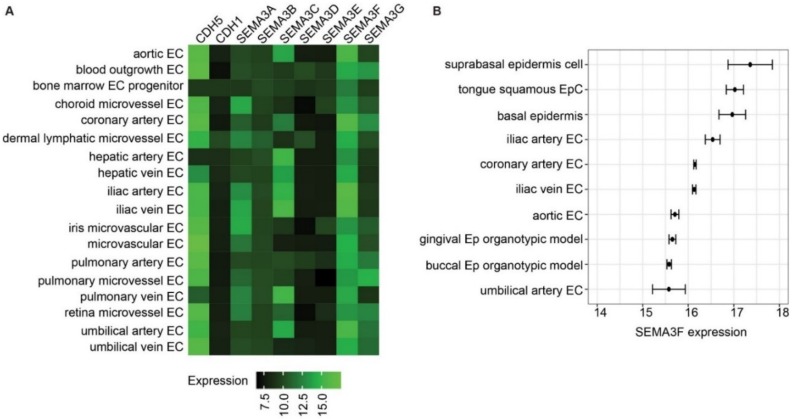
High expression of SEMA3F in various types of endothelial cells. (**A**) Heatmap of Class III semaphorin expression of all types of human endothelial cells profiled on Affymetrix Human Genome U133 Plus 2.0 Array platform (GEO accession: GPL570). Numbers indicate normalized microarray signals on log2 scale. Black color: lower expression and green color: higher expression. Expression signals of CDH5 and CDH1 were included as references. (**B**) SEMA3F expression in 10 human cell types with highest SEMA3F expression. Data is presented as mean ± SEM; *n* = 3.

**Figure 2 ijms-21-01471-f002:**
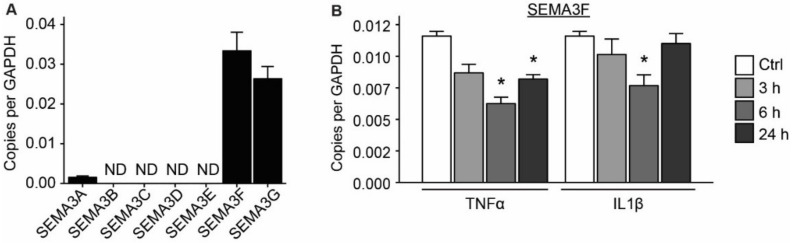
Confirmation of SEMA3F expression in human umbilical vein endothelial cells (HUVECs) and its response to inflammatory factors. (**A**) Quantification of expression of Class III semaphorins with qPCR in HUVECs. Results are presented as copy numbers per GAPDH, mean ± SEM, *n* = 3. (**B**) SEMA3F expression in HUVECs stimulated with the inflammatory factors, TNFα (10 ng/mL) or IL1β (20 ng/mL). RNA was harvested 3, 6 and 24 h after stimulation started. Results are presented as copy numbers per GAPDH, mean ± SEM, *n* = 3, * *p* < 0.05.

**Figure 3 ijms-21-01471-f003:**
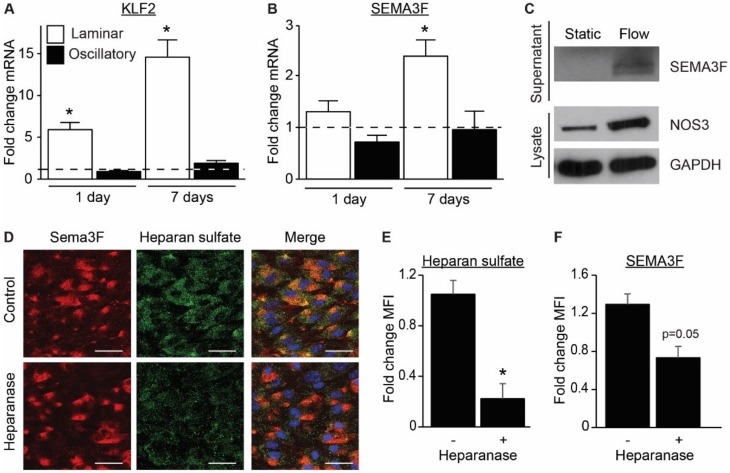
Regulation of SEMA3F expression and secretion by shear stress. (**A**,**B**) Quantitative qPCR analysis of KLF2 (**A**) and SEMA3F (**B**) mRNA isolated from HUVECs cultured under laminar or oscillatory flow conditions (10 dyn/cm^2^) for 1 or 7 days compared to static culture conditions. Results are presented relative to static cultured cells, set as 1, dotted line. Mean ± SEM of *n* = 3, * *p* < 0.05. (**C**) Immunoblot analysis of SEMA3F protein dimer in the supernatant and NOS3 or GAPDH in cell lysates. Representative of *n* = 2–3. (**D**–**F**) Immunofluorescence staining of SEMA3F (red), Heparan sulfate (green) and nuclei (blue) in HUVECs cultured under laminar flow conditions for 7 days and treated with or without heparanase (1.5 µg/mL) for 2 hours at the end of the 7 days culture. Scale bar 40 µm. Quantification of mean fluorescence intensity of heparan sulfate (**E**) and SEMA3F (**F**) of 3 independent experiments. Mean ± SEM, * *p* < 0.05.

**Figure 4 ijms-21-01471-f004:**
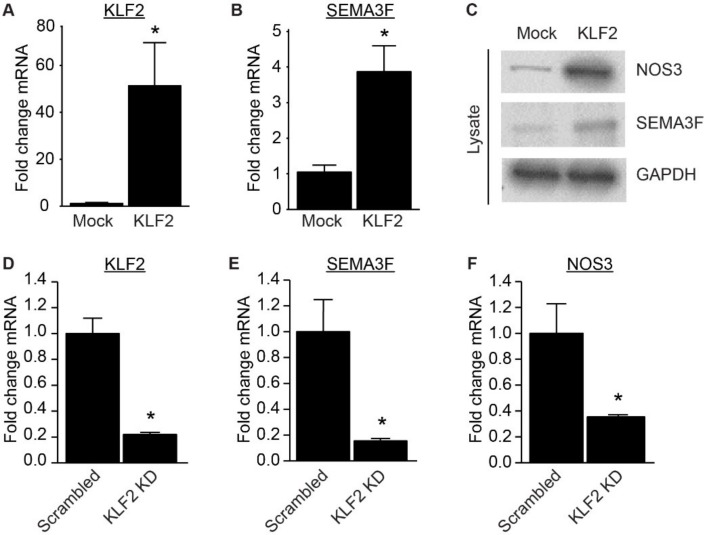
SEMA3F is induced by KLF2. (**A**,**B**) Quantitative qPCR analysis of KLF2 (**A**) and SEMA3F (**B**) mRNA isolated from HUVECs transduced with mock or KLF2 overexpressing lentivirus. Results are presented relative to cells transduced with mock virus, set as 1. Mean ± SEM of *n* = 3; * *p* < 0.05. (**C**) Representative immunoblot analysis of NOS3, SEMA3F or GAPDH in protein lysates of HUVEC transduced with mock or KLF2 overexpressing lentivirus. (**D**–**F**) KLF2 (**D**), SEMA3F (**E**) and NOS3 (**F**) mRNA expression in HUVECs after transduction with lentiviral anti-KLF2 shRNA (KLF2 KD) or scrambled shRNA (scrambled) and cultured under laminar flow conditions (10 dyn/cm^2^) for 7 days compared to static culture conditions. Results are presented relative to scrambled, set as 1. Mean ± SEM of *n* = 3, * *p* < 0.05.

**Figure 5 ijms-21-01471-f005:**
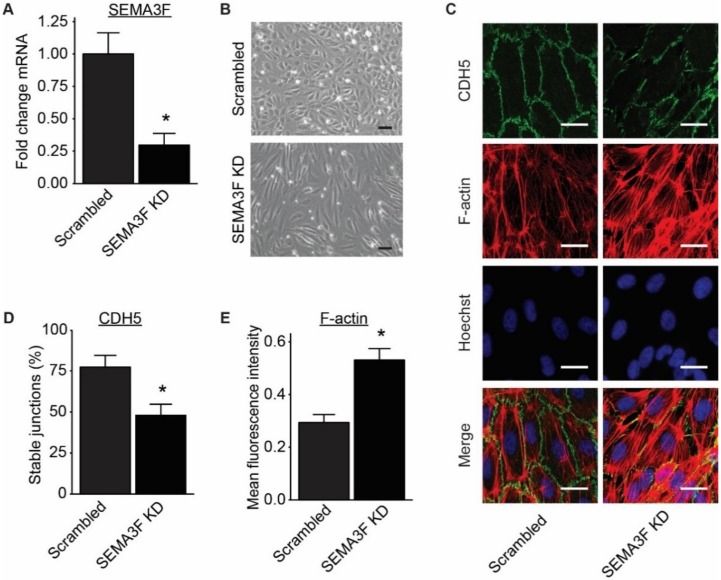
Reduction of SEMA3F expression in endothelial cells leads to disrupted endothelial adherens junctions. (**A**) SEMA3F mRNA expression after transduction with lentiviral anti-SEMA3F shRNA (SEMA3F KD) or scrambled shRNA (Scrambled). Mean ± SEM of *n* = 3, * *p* < 0.05. (**B**) Bright field microscope images from HUVECs transduced with anti-SEMA3F shRNA (SEMA3F KD) or scrambled shRNA (scrambled). Scale bar 40 µm. (**C**–**E**) Immunofluorescence staining of CDH5 (green), F-actin (red) and nuclei (blue) in SEMA3F KD or scrambled HUVECs. Scale bar 10 µm (**D**,**E**) Bar graphs show the quantification of percentage of stable adherens junctions (**D**) and F-actin staining (**E**). Mean ± SEM of *n* = 3, * *p* < 0.05.

**Figure 6 ijms-21-01471-f006:**
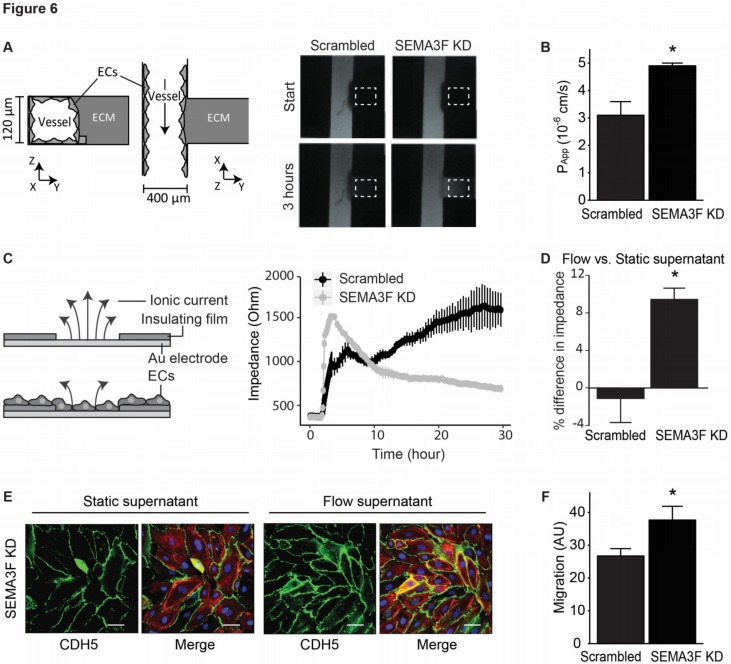
Reduction of SEMA3F expression impairs barrier function of the endothelial monolayer. (**A**,**B**) HUVECs transduced with anti-SEMA3F shRNA (SEMA3F KD) or scrambled shRNA (scrambled) were seeded in the vertical channel of a microfluidic device with collagen in the horizontal channel. At the start of the measurement the fluorescently labeled albumin were infused in the vessel lumen and after 30 minutes albumin leakage outside the vessel channel was visualized and quantified. Dotted box indicates region of quantification. Bar graph (**B**) shows quantification of permeability presented as mean ± SEM of *n* = 3, * *p* < 0.05. (**C**) Transendothelial electrical resistance of HUVECs transduced with anti-SEMA3F shRNA (SEMA3F KD) or scrambled shRNA (scrambled) and seeded on electric cell–substrate impedance sensing (ECIS) electrodes. Data are from one experiment representative of five independent experiments (mean ± SD of triplicate samples). (**D**) HUVECs transduced with anti-SEMA3F shRNA (SEMA3F KD) or scrambled shRNA (scrambled) were seeded and cultured on ECIS electrodes in the presence of supernatant of non-transduced endothelial cells cultured under static or laminar flow (7 days, 10 dyn/cm^2^). Results are presented as percentage difference in impedance of the cells treated for 42 hours with supernatant from flow compared with static endothelial cultures. Data are mean ± SD of *n* = 2–3, * *p* < 0.05. (**E**) Immunofluorescence staining of CDH5 (green), F-actin (red) and nuclei (blue) in SEMA3F KD HUVECs cultured in the presence of supernatant of non-transduced endothelial cells cultured under static or laminar flow (7 days, 10 dyn/cm^2^) for 24 h. Scale bar 50 µm. (**F**) Migration of THP-1 cells across SEMA3F KD or scrambled HUVEC-coated filters towards MCP1 (10 µM). Data are mean ± SEM of *n* = 3, * *p* < 0.05.

**Figure 7 ijms-21-01471-f007:**
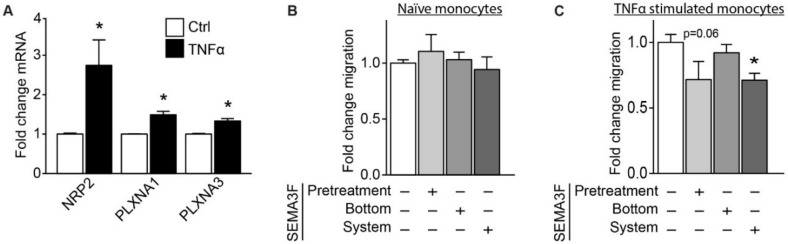
SEMA3F inhibits migration of TNFα stimulated monocytes. (**A**) Quantitative qPCR analysis of NRP2, PLXNA1 and PLXNA3 mRNA in THP-1 cells stimulated for 15 h with TNFα (2 ng/mL), presented relative to mRNA in unstimulated cells (control), set as 1. Mean ± SEM of *n* = 3, * *p* < 0.05. (**B**,**C**) Migration of THP1 cells unstimulated (**B**) or stimulated for 15 h with TNFα (2 ng/mL, **C**) and with or without pretreatment with SEMA3F (500 ng/mL) or exposure to SEMA3F (500 ng/mL) in the bottom or in the whole Transwell system towards MCP1 (10 µM). Results are presented relative to cells not exposed to SEMA3F, set as 1. Mean ± SEM of *n* = 3, * *p* < 0.05.

**Table 1 ijms-21-01471-t001:** List of primers used.

Gene	Forward Sequence	Reverse Sequence
*GAPDH*	CCTGCACCACCAACTGCTTA	GGCCATCCACAGTCTTCTGAG
*SEMA3A*	CTTGCATTCATCTCTTCTGGTGT	GTGCCAAGGCTGAAATTATCCT
*SEMA3B*	ACATTGGTACTGAGTGCATGAAC	GCCATCCTCTATCCTTCCTGG
*SEMA3C*	GTATGTCTGTGGGAGTGGCG	ACGTTGGGGTTGAAAGAGCA
*SEMA3D*	TCAGAGCACTACTGGCTCAAT	ATCGGAGGTACTGCCTTCTTG
*SEMA3E*	TGTCCTTTTGACCCCAGCTC	CGTCATGCTCAGTGCGGATA
*SEMA3F*	CCACAGCGCATCGAGGAAT	CATGGGGTTGTAGGCACCTG
*SEMA3G*	TGGCTCGAACCATGTCACTG	CATTTGTTCACCAGCACCCG
*KLF2*	CTACACCAAGAGTTCGCATCTG	CCGTGTGCTTTCGGTAGTG
*NOS3*	TGATGGCGAAGCGAGTGAAG	ACTCATCCATACACAGGACCC
*NRP2*	GCTGGCTATATCACCTCTCCC	TCTCGATTTCAAAGTGAGGGTTG
*PLXNA1*	CGTGCTGTTCACTGTGTTCG	ACTGGATGCGCTCCTTAATCT
*PLXNA3*	CGGACATGTTCAGTCTCGTGTA	CGCTGACGAAGCCGTAGAT

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
