# Peer review of "Endothelial Semaphorin 3F Maintains Endothelial Barrier Function and Inhibits Monocyte Migration"

_ijms, 2020, doi:10.3390/ijms21041471_

Round 1

Reviewer 1 Report

Semaphorin 3F (SEMA3F) is a member of the class 3 semaphorins involved in axon guidance, angiogenesis, tumor progression and immune response. In this manuscript, Zhang and colleagues find that SEMA3F expression is reduced by inflammatory stimuli, on the other hand, SEMA3F is induced by laminar flow. They also show that secreted SEMA3F by laminar flow binds to heparan sulfates on the surface of ECs. In addition, they show that SEMA3F inhibits activated monocyte migration. The authors describe endothelial SEMA3F as a mediator of EC quiescence. This is well-written manuscript supported by in vitro clear-cut experiments, for this reason I suggest only minor revision.

Minor concerns:

The main finding in this manuscript is that endothelial SEMA3F expression is regulated by inflammatory cytokines and shear stress, however, is this SEMA3F specific in the class3 semaphorine family? By providing this data, the authors would significantly improve the general impact of this study.

In the fig3, the authors show the KLF2-dependent regulation of SEMA3F under laminar shear stress. To confirm this, knockdown of KLF2 in HUVEC would suppress the induction of SEMA3F by laminar flow.

Adding exogenous SEMA3F would rescue disrupted endothelial adherens junctions (fig. 4) and impaired barrier function (fig. 5) by SEMA3F knockdown. These data exclude off-target effect of shRNA and support the conclusion of this manuscript.

Author Response

We would like to thank the reviewer for her/his thoughtful comments regarding our study. In response to the comments raised we have made appropriate revisions to the manuscript, figures, and supplementary materials, and provide specific responses to each minor concern raised.

The main finding in this manuscript is that endothelial SEMA3F expression is regulated by inflammatory cytokines and shear stress, however, is this SEMA3F specific in the class3 semaphorine family? By providing this data, the authors would significantly improve the general impact of this study.

The expression of the lower expressed SEMA3A is also regulated by inflammatory cytokines. We agree with the reviewer that adding the analysis of other expressed SEMA3 class member upon inflammatory cytokine stimulation or shear stress would significantly improve the general impact of our study. We have now analyzed the expression levels of the other SEMA3 class members we detected in the endothelial cells treated by inflammatory cytokines or shear stress, and added these to the revised manuscript as Supplemental figure 1A-B.

In the fig3, the authors show the KLF2-dependent regulation of SEMA3F under laminar shear stress. To confirm this, knockdown of KLF2 in HUVEC would suppress the induction of SEMA3F by laminar flow.

As requested, we generated KLF2 knock down HUVEC to confirm the KLF2-dependent regulation of SEMA3F under laminar shear stress. Our results show that knockdown of KLF2 in endothelial cells suppressed the induction of SEMA3F in cells cultured under laminar flow conditions. The results are presented in new Figure 4 of our revised manuscript.

Adding exogenous SEMA3F would rescue disrupted endothelial adherens junctions (fig. 4) and impaired barrier function (fig. 5) by SEMA3F knockdown. These data exclude off-target effect of shRNA and support the conclusion of this manuscript.

In response to the reviewer’s comments we present new data in the Figure 6. We demonstrate that treating SEMA3F KD endothelial cells with supernatant from laminar flow cultured endothelial cells increases CDH5 localization at adherence junctions (Figure 6E). In addition, supernatant from laminar flow cultured endothelial cells increased the endothelial barrier, compared to supernatant form static cultured endothelial cells (Figure 6F).

Reviewer 2 Report

The paper by Zhang and coworkers describes the presence of semaphorins in endothelial cells and role of SEMA3F in maintaining barrier function of vascular endothelium in laminar flow condition and its downregulation under inflammatory stimuli, thus favoring monocyte migration.

The paper is well written and results sound.

Author Response

We would like to thank the reviewer for her/his thoughtful comment regarding our study, and are pleased that the reviewer thought the manuscript was well-written with sound results.  

Round 2

Reviewer 1 Report

N/A